# Head-Mounted Display-Based Microscopic Imaging System with Customizable Field Size and Viewpoint [note 1]

**DOI:** 10.3390/s20071967

**Published:** 2020-04-01

**Authors:** Tadayoshi Aoyama, Sarau Takeno, Masaru Takeuchi, Yasuhisa Hasegawa

**Affiliations:** 1Department of Micro-Nano Mechanical Science and Engineering, Nagoya University, Furo-cho 1, Chikusa Word, Nagoya 464-8603, Japan; takeno@robo.mein.nagoya-u.ac.jp (S.T.); takeuchi@mein.nagoya-u.ac.jp (M.T.); yasuhisa.hasegawa@mae.nagoya-u.ac.jp (Y.H.); 2Japan Science and Technology Agency, PREST, 4-1-8 Honcho, Kawaguchi, Saitama 332-0012, Japan

**Keywords:** HMD, micro-macro interface, view-expansive microscope

## Abstract

In recent years, the use of microinjections has increased in life science and biotechnology fields; specific examples include artificial insemination and gene manipulation. Microinjections are mainly performed based on visual information; thus, the operator needs high-level skill because of the narrowness of the visual field. Additionally, microinjections are performed as the operator views a microscopic image on a display; the position of the display requires the operator to maintain an awkward posture throughout the procedure. In this study, we developed a microscopic image display apparatus for microinjections based on a view-expansive microscope. The prototype of the view-expansive microscope has problems related to the variations in brightness and focal blur that accompany changes in the optical path length and amount of reflected light. Therefore, we propose the use of a variable-focus device to expand the visual field and thus circumvent the above-mentioned problems. We evaluated the observable area of the system using this variable-focus device. We confirmed that the observable area is 261.4 and 13.9 times larger than that of a normal microscope and conventional view-expansive microscopic system, respectively. Finally, observations of mouse embryos were carried out by using the developed system. We confirmed that the microscopic images can be displayed on a head-mounted display in real time with the desired point and field sizes.

## 1. Introduction

In 1980, the first study on a transgenic mouse created by injecting a foreign gene into the pronuclei of fertilized mouse oocytes was reported [1]. In 1989, a study on a knockout mouse lacking a specific endogenous gene was reported [2]. These genetically modified animals are not only applied for the functional analysis of genes, but also for disease state analysis and therapeutic drug development, as disease model animals. In vitro fertilization is regarded as an important technique for the treatment of human infertility, and the efficient production of livestock [3,4]. Production of genetically modified animals and in vitro fertilization of livestock are mainly performed using the microinjection technique. Details of the microinjection technique are described below. To efficiently microinject a large number of embryos, the workspace is divided into three areas, as shown in Figure 1, so as not to confuse the pre-injected embryo and the injected embryo. The pre-injected embryos and injected embryos are developed in Workspaces 1 and 3, respectively, and microinjection is performed in Workspace 2. Therefore, when injecting, it is necessary to move the pre-injected embryo from Workspace 1 to Workspace 2, and to move the injected embryo Workspace 2 to Workspace 3. When transporting embryos, a wide field of view is required to confirm the intended transport and embryo positions. However, at the time of microinjection, because accurate injection into the pronucleus is critical, observation at a high magnification (high-resolution) is required. Thus, both a large field of view and a high-resolution observation is required for this process. Each time, the objective lens must be changed, and the aperture must be adjusted. Having to perform these tasks while maintaining an uncomfortable position for an extended period of time, i.e., to view the display, places a significant burden on the operator. As described above, when microscale operations are performed, visual information is critical. However, limited visual information and a painful postural stance makes the operation even more difficult for the operator. Moreover, fluctuations in the success rate exist because of human fatigue and a low level of reproducibility, meaning that accuracy, reproducibility, and productivity are problems.

In this study, we developed a microscopic image presentation apparatus based on the concept of view-expansive microscope [5] to improve the efficiency of the microinjection process, and to reduce the burden on the operator. With the proposed apparatus, high resolution and large fields of view are feasible owing to the use of a view-expansive microscope. The apparatus also reduces the burden on the operator because it includes a head-mounted display (HMD) and virtual reality (VR) controllers. The earliest version of the apparatus was introduced in [6]. In this study, the view-expansive area of the proposed image presentation apparatus was evaluated, and the observable range was verified. Then, we developed a system that allows enlarged images with an expanded field of view to be viewed by the operator, regardless of the desired position. Finally, the effectiveness of the system was verified through an experiment that involves the observation of mouse embryos.

## 2. Related Works

When using an optical lens microscope, although it is possible to maintain a high resolution with low magnification, the microscope and objective lens need to be enlarged, which is substantially expensive. Generally, the field of view and resolution of a microscope have an inverse relationship, and numerous studies have been performed to realize a wide field of view and high resolution. Chow et al. expanded the visual field by using a moving stage to capture images from multiple viewpoints, and subsequently processing the images [7]. However, moving the stage at a high speed is difficult, and expanding the visual field is a time-consuming process. Benjamin et al. developed a microscope that can select viewpoints at high speeds because of a dual-axis mirror that has been inserted between the objective lens and tube lens [8]. With their system, the light is distorted as it passes through the outer edge of the objective lens; thus, high resolution can be maintained by using a deformable mirror. It requires a huge objective lens and an adaptable deformable mirror; thus, their system is considerably cost-expensive and lacks versatility. Lecoq et al. developed a two-axis microscope with two separate scan paths and two articulated arms [9]; however, the observable position is fixed. This makes it difficult to continuously photograph positions. Terada et al. developed a system that can capture different viewpoints in 10 ms; they developed a novel optical system and inserted it into the space between the objective lens and observation target [10]. However, the range of positions at which the viewpoint can be changed is limited, and the field of view is inadequate.

The recent development of lasers and imaging devices has encouraged the development of various digital holographic microscopes (DHMs). DHMs record holograms formed by a reference light and object light using an imaging device; the phase and amplitude information of microscopic objects are then reconstructed. With DHMs, a wide-field 3D image can be obtained with a single shot. However, DHMs have problems related to the resolution and time required for image reconstruction. For instance, in-line DHMs yield reproduced images with low resolution owing to the influence of the transmitted zero-order light and conjugate image. It is possible to increase the resolution by using a synthetic aperture and super-resolution pixel technology, but multiple images must be captured [11,12]. For this reason, off-axis and phase-shifting holographic microscopes have been developed. However, because of pixel size limitations, off-axis DHMs must have a configuration that is similar to that of in-line holographic microscopes; moreover, it is difficult to remove all of the noise [13,14]. Alternatively, phase-shifting holographic microscopes require that a single image be captured multiple times; this means that the process of obtaining numerous holograms is substantially time expensive [15]. Therefore, Tahara et al. developed parallel phase-shifting digital holographic microscopy; their system yields reconstructed images with a higher resolution than those achievable via one-shot inline holography [16]. However, a special optical element is required for their system, and information on the delay of each phase in the interference fringes is insufficient.

Some researchers have attempted to develop systems that display reconstructed images in real time by improving the image acquisition speed and utilizing GPUs and CPUs to realize high-speed image processing [17,18,19,20,21]. Kim et al. proposed a high-speed tomography system that uses a GPU to enable high-speed image acquisition and parallel processing; consequently, measurement and visualization of 96${}^{3}$ voxels were achieved in 1.3 s [17,18]. Dardikman et al. improved this system, making 3D reconstruction faster than 25 frames/s (fps) for 256 × 256 pixel interferograms with 73 different projection angles [19]. Natan et al. used a CPU and GPU to achieve a video frame rate that exceeds 35 fps for a hologram of 2048 × 2048 [20,21]. However, because the processing time increases as the number of hologram pixels increases, it is difficult to preserve this video frame rate under the condition of a wider field of view. A semi-micromanipulator with optoelectronic tweezers and a holographic microscope has also been developed [22]; the system is not suitable for microinjection that requires real-time image presentation with high-resolution images.

Thus far, as described above, the microscopic imaging systems that achieve both high-resolution and wide-area imaging are not sufficient as an image presentation interface for improving microinjection. Our research group has proposed a view-expansive microscope system that uses high-speed vision to allow images to be captured as the viewpoint is shifted using a galvanometer mirror [5]. The field of view can be simply expanded by installing a mirror behind a commercially available objective lens. We also improved the field of view of the view-expansive microscope by using a variable-focus device [23]. We then employed the microscope system to track multiple mobile microorganisms. However, the viewpoint range of the microscope system described in [23] is limited to one direction because the system was designed for a microchannel. Therefore, in this study, we expanded on the idea proposed in [23] to create a 2D view-expansion system; we also developed a microscopic image presentation apparatus for effective microinjection.

## 3. Microscopic Image Presentation Apparatus

Figure 2 illustrates the configuration of the proposed microscopic image presentation apparatus. The proposed apparatus uses the expanded-view microscopic imaging system to output a magnified and wide-area image when applied in a microscopic environment [5]. Figure 3 shows an overview of the proposed image presentation apparatus. The proposed apparatus incorporates a simplified microscope unit (KTL-N21B-1, Kyowa Optical Co.), an upright microscope (CX41, OLYMPUS), an objective lens (×10, Kyowa Optical Co.), a two-degrees-of-freedom galvano mirror (6210 HSM 6 mm, 532 nm, Cambridge Technology), a high-speed camera (MQ003CG-CM, Ximea), a control computer (Windows 7 Professional 64-bit OS, Hewlett-Packard 212B, Intel Xeon CPU E5-1603 2.80 GHz, DDR2 8 GB), a D/A board (PCX-340416, Interface), a variable-focus device, an HMD (Oculus rift, Oculus), and a VR controller (Oculus Touch, Oculus). The developed system can capture images as large as 1920 pixels × 1440 pixels (1.42 mm × 1.07 mm). The working distance (WD) and numerical aperture of the objective lens are 33 mm and 0.3, respectively. Dielectric multilayer reflecting mirrors which reflect light of wavelength 532 nm are used in this system. The resolution of this microscopic system is approximately 1.08 μm defined based on the Rayleigh criterion. Movement of the galvano mirror is dependent on the angle of the head of the operator and the input to the VR controller. According to the instructions input to the VR controller, the viewing area can vary between ×1 (0.47 mm × 0.36 mm) and ×9 (1.42 mm × 1.07 mm), and the image is resized to 640 × 480 pixels to adjust HMD display. The system can capture one viewpoint image and display it on a HMD within 6 ms; thus, the system requires 6*N* ms to capture and display a *N* times expanded image.

Figure 4 shows the optical path between the objective lens and target object on the variable-focus device. $l_{1}$, $l_{2}$, and $l_{3}$ indicate the optical paths between the objective lens and pan-axis mirror, between the pan-axis mirror and tilt-axis mirror, and between the tilt-axis mirror and target object on the variable-focus device, respectively. $l_{m}$ indicates the distance between the pan axis and tilt axis. α and β indicate the optical angles of the pan-axis mirror and tilt-axis mirror, respectively.

The optical path from the objective lens to the target object must always be equal to WD; hence, we derived the shape of the variable-focus device as follows: $l_{1}$ and $l_{2}$ are constant, because the installation conditions $l_{2}$ and $l_{3}$ become
(1)$$\begin{array}{c} l_{2}=\frac{l_{m}}{\cos\alpha },\;\mathrm{and} \end{array}$$
(2)$$\begin{array}{c} l_{3}=WD-l_{1}-l_{2}. \end{array}$$

Therefore, in an orthogonal coordinate system in which the top of the objective lens is the origin, the coordinates of the set of points on the variable-focus device are expressed as follows: (3)$$\begin{array}{cc} x & =l_{1}-l_{2}\sin\alpha -z\tan\alpha , \end{array}$$(4)$$\begin{array}{cc} y & =l_{m}+z\tan\beta ,\;\mathrm{and} \end{array}$$(5)$$\begin{array}{cc} z & =\frac{l_{3}}{\sqrt{1+{\tan\alpha }^{2}+{\tan\beta }^{2}}}. \end{array}$$

In this study, we designed a variable-focus device, as shown in Figure 5, according to the conditional expressions in Equations (Equation 3)–(5).

## 4. Evaluation of Captured Microscopic Image Using the View-Expansive Microscope System

In this section, we demonstrate the effectiveness of the variable-focus device through the analysis of the microscopic images and comparison of the ranges in which clear images can be acquired under the condition that the target object on the plane is photographed on the variable-focus device.

### 4.1. Evaluation Index

#### 4.1.1. Average Brightness

The average luminance value $I_{aver}$ can be derived from the luminance value I(x,y), as follows:(6)$$\begin{array}{c} I_{ave}=\frac{1}{N_{x}N_{y}}\sum_{x,y}I\left(x,y\right). \end{array}$$
where $N_{x}$ and $N_{y}$ are the numbers of pixels in the x and y directions, respectively.

#### 4.1.2. Average Edge Intensity

The edge intensity at position (x,y) can be computed by using the gradient value, as follows:(7)$$\begin{array}{cc} \; & E\left(x,y\right)=\sqrt{{\left\{I\left(x+1,y\right)-I\left(x,y\right)\right\}}^{2}+{\left\{I\left(x,y+1\right)-I\left(x,y\right)\right\}}^{2}}. \end{array}$$
The average edge intensity can then be derived as follows:(8)$$\begin{array}{c} E_{ave}=\frac{1}{N_{x}N_{y}}\sum_{x,y}E\left(x,y\right). \end{array}$$

### 4.2. Evaluation of Captured Images

We conducted an experiment in which we photographed a retroreflective sheet on the plane and variable-focus device, as shown in Figure 6. We acquired a total of 10,201 images by adjusting the pan and tilt mirror angles in increments of $0.2^{\circ }$ within the range of $-10^{\circ }$ to $10^{\circ }$ after capturing each image. We then evaluated the variations in brightness and focal blur. The photography-based experiments were repeated 10 times to obtain the average brightness $I_{ave}$ and edge intensity $E_{ave}$ values. We used a different retroreflective sheet in every experiment in consideration of the influence of the size and number of glass beads affixed to the sheets.

Figure 7, Figure 8 and Figure 9 show composites of the captured images, the colormap of the average luminance values with respect to the pan- and tilt-axis mirror angles, and the colormap of the average normalized edge intensity values with respect to the pan- and tilt-axis mirror angles, respectively. The maximum average edge intensity was set to 100%, as shown in the colormap of the average edge intensity. The maximum average brightness for the pan- and the tilt-axis mirror angles was set to 100%, as shown in the colormap of the average brightness.

Figure 8 shows that, during observation of the retroreflective sheet on the variable-focus device, the pan-axis brightness remained nearly constant, whereas the tilt-axis brightness decreased, as the mirror angle deviated from the center. This occurred because the light reflected by the tilt-axis mirror began expanding beyond the pan-axis mirror surface as the tilt-axis mirror angle increased. Figure 7 and Figure 8 show that, during observation of the retroreflective sheet on the plane, the brightness value decreased as the mirror angle deviated from the center because of the influence of the focal blur.

Figure 9 shows that, during observation of the retroreflective sheet on the variable-focus device, as the mirror angle deviated from the center, the edge intensity along the pan axis remained almost constant, whereas the brightness value along the tilt axis decreased. Focal blur occurred when the retroreflective sheet was observed on the plane, and as the pan- and tilt-axis mirror angles deviated from the center. These results indicate that the variations in brightness and focal blur were minimized as a result of using the variable-focus device.

### 4.3. Comparison of Observable Area Results

In this section, we describe the observable area, which corresponds to the area in which clear images can be captured without any focal blur. Note that an image photographed in the depth of field is referred to as a clear image. The depth of field includes the front depth of field $d_{f}$ and rear depth of field $d_{b}$. These values can be derived by using the distance between the objective lens and target object *L*, the focal length *f*, the F number *F*, and the permissible circle of confusion, as follows:(9)$$\begin{array}{cc} d_{f} & =\frac{\epsilon LF(L-f)}{f^{2}+\epsilon F\left(L-f\right)},\;\mathrm{and} \end{array}$$(10)$$\begin{array}{cc} d_{b} & =\frac{\epsilon LF(L-f)}{f^{2}-\epsilon F\left(L-f\right)}. \end{array}$$

In this system, L=33 mm, f=20 mm, and ϵ=7.4
μm. Thus, according to Equations (Equation 9) and (10), the depths of focus were $d_{f}=13.2$
μm and $d_{b}=13.2$
μm, respectively.

Accordingly, the value of the edge intensity of the point moved by ±13
μm in the direction of the optical axis, relative to the position at which the edge intensity reached a maximum, was identified as the threshold value. We defined the range in which the average edge intensity was larger than the threshold value to be the observable area. The threshold value was calculated as the mean value of the edge intensities of the 10 captured images. Figure 10 shows the images with edge intensities do not exceed the threshold value. These images show that focal blur did not occur when the edge intensity did not exceed the threshold value. Figure 11 shows the color map of the observable area that satisfies the threshold values of the normalized edge intensity. It can be seen that the range of the photographed curved surface was wider than that of the planar surface.

The observable areas of all surfaces are provided in Table 1 with respect to the visual field of a normal microscopic image.

## 5. Implemented Algorithm

### 5.1. Angle Control of Mirror: Viewpoint Adjustment

In the proposed system, the desired pan- and tilt-axis mirror angles, $\theta_{p}^{d}$ and $\theta_{t}^{d}$, respectively, have been defined as follows by using the input to the VR controller and the head angle of the operator:(11)$$\begin{array}{cc} \theta_{p}^{d} & =\theta_{p}+kI_{cp}+mI_{hp}, \end{array}$$(12)$$\begin{array}{cc} \theta_{t}^{d} & =\theta_{t}+lI_{ct}+nI_{ht}, \end{array}$$
where $I_{cp}$ and $I_{ct}$ are inputs to the VR controller, $I_{hp}$ and $I_{ht}$ are inputs to the HMD, and *k*, *l*, *m*, and *n* are constants.

### 5.2. Creation of the View-Expanded Image of Desired Size

(a) Initial positions 

The initial angles of the pan- and tilt-axis mirror, i.e., $\theta_{p}^{s}$ and $\theta_{t}^{s}$, respectively, can by derived by using the visual field enlargement ratio, as follows:(13)$$\begin{array}{cc} \theta_{p}^{s} & =\theta_{p}^{d}+uS_{x}\frac{a-1}{2}, \end{array}$$(14)$$\begin{array}{cc} \theta_{t}^{s} & =\theta_{t}^{d}+vS_{y}\frac{a-1}{2}, \end{array}$$
where $S_{x}$ and $S_{y}$ are the length and width, respectively, of the real field of view when X=1. *u* and *v* are constants, and *a* is an integer that satisfies the following condition:(15)$$\begin{array}{c} {\left(a-1\right)}^{2}<X\leq a^{2}\;\;\left(a=1,2,3\right), \end{array}$$
where *X* is the visual field enlargement ratio.

(b) Image acquisition 

Images were acquired by changing the pan- and tilt-axis mirror angles ($\Delta \theta_{p}$ and $\Delta \theta_{t}$) for each image-capture experiment in the order shown in Figure 12.
(16)$$\begin{array}{cc} \Delta \theta_{p} & =\frac{S_{x}}{l_{2}+l_{3}},\;\mathrm{and} \end{array}$$
(17)$$\begin{array}{cc} \Delta \theta_{t} & =\frac{S_{y}}{l_{3}}. \end{array}$$

(c) HMD display 

After acquiring the image by following the above-described image acquisition method, the resulting rectangular image was cut such that the length and width relative to the viewpoint center were $S_{x}\sqrt{X}\times S_{y}\sqrt{X}$ for pan- and tilt-axis mirror angles of $\theta_{p}^{d}$ and $\theta_{t}^{d}$, respectively. The image was resized to 640 × 480 pixels; then, the composed image was displayed on the HMD as shown in Figure 13.

## 6. Observation Experiment

For observation, we placed mouse embryos on top of a 1-mm polydimethylsiloxane (PDMS) sheet that was positioned on the variable-focus device. During this observation, the operator did not touch the microscope, but controlled the viewpoint and visual field enlargement ratio by adjusting their head angle, or the VR controller. Figure 14 shows an experimental scene and Figure 15 shows the images that were acquired when the visual field enlargement ratio was set to ×1, ×4, and ×9. Thus, we have confirmed that the proposed system can display microscopic images of the desired viewpoint and visual field in real time.

## 7. Conclusions

In this study, we minimized brightness and focal blur variation of the previous view-expansive microscope, and developed a microscopic image presentation apparatus that can display the desired visual field and viewpoint of an image on a HMD in real time.

We demonstrated that it is possible to observe an area that is as much as 261.4 and 13.9 times larger than what is feasible with a normal microscope and planar view-expansive observation, respectively. We confirmed that the proposed system can expand the desired viewpoint by a factor of 1–9 in real time. Using the proposed system, the information contained within images with a wide visual field that is required by a manipulator can be extracted with high resolution. Thus, operations that were limited by an undesirably small field of view because of the trade-off with resolution (e.g., micro-object rotation, position adjustment, and microinjection) can be performed under less restrictive conditions. Future work includes the development of a microinjection system that uses the proposed image presentation apparatus and its evaluation. 

## Figures and Tables

**Figure 1 sensors-20-01967-f001:**
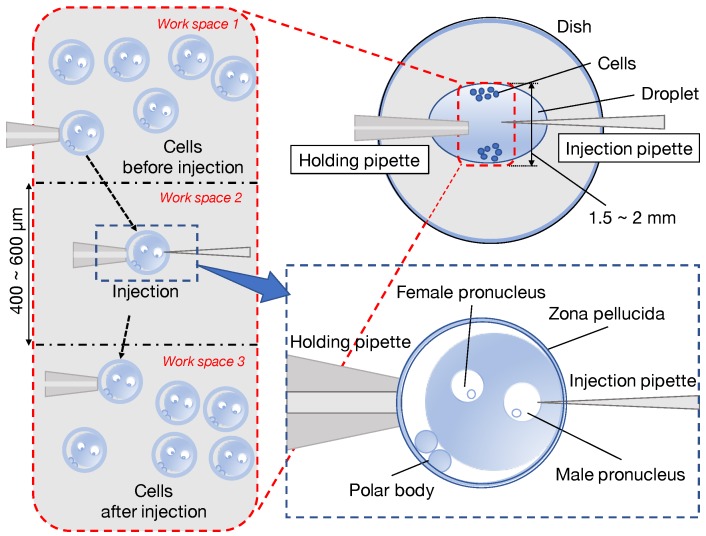
Workspaces for microinjection process.

**Figure 2 sensors-20-01967-f002:**
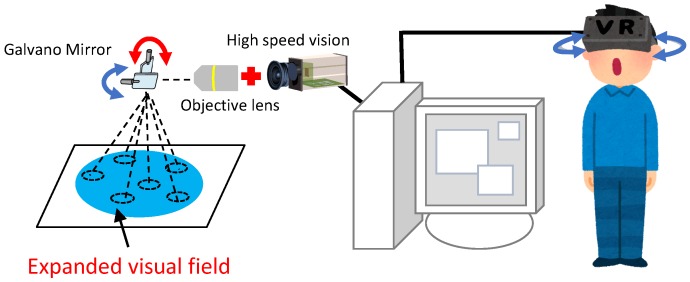
Configuration of the proposed microscopic image presentation apparatus.

**Figure 3 sensors-20-01967-f003:**
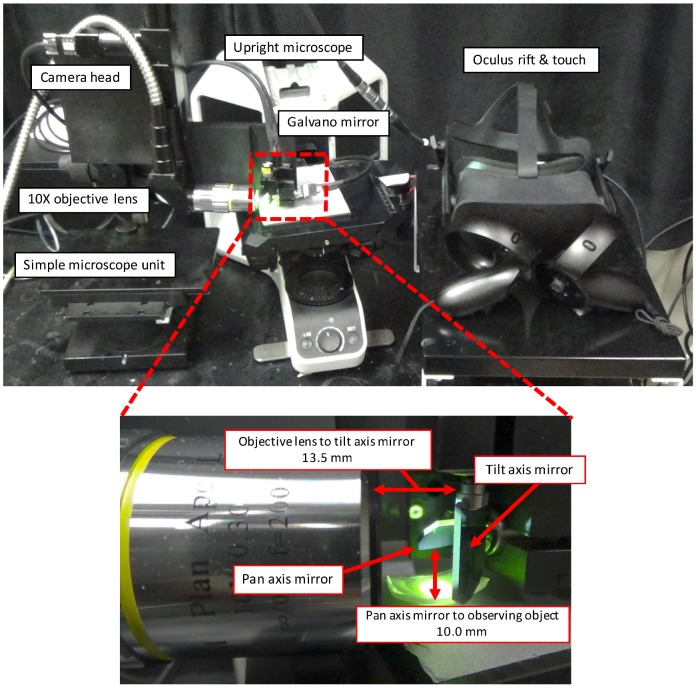
Overview of the proposed image presentation apparatus.

**Figure 4 sensors-20-01967-f004:**
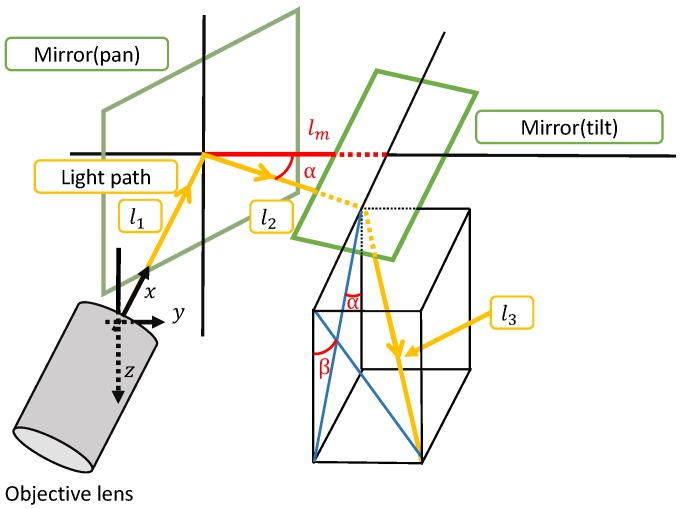
Optical path between the objective lens and target object on the variable-focus device.

**Figure 5 sensors-20-01967-f005:**
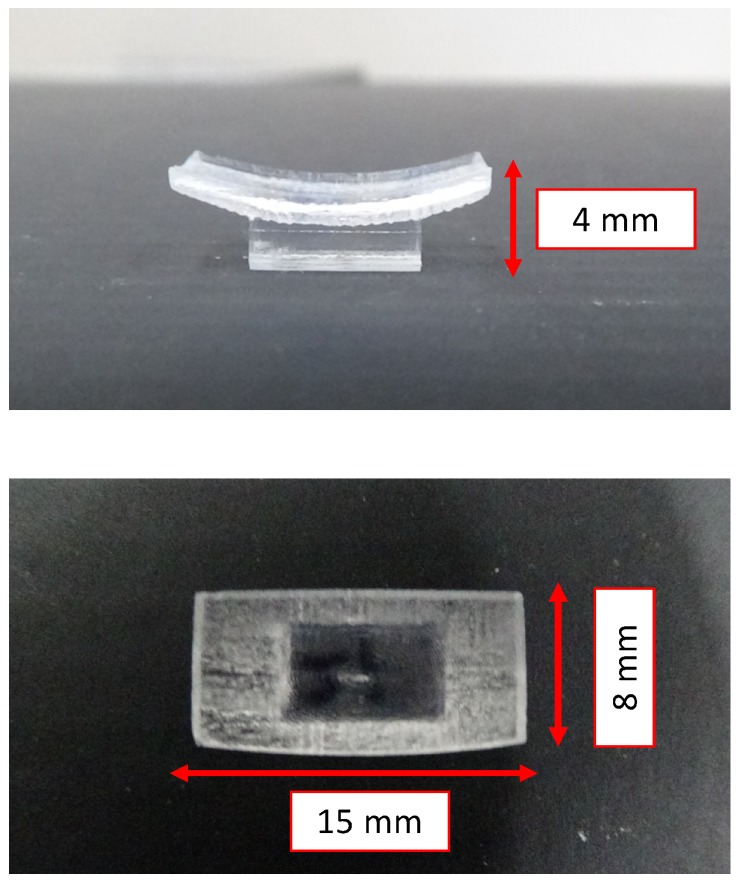
Overview of the variable-focus device.

**Figure 6 sensors-20-01967-f006:**
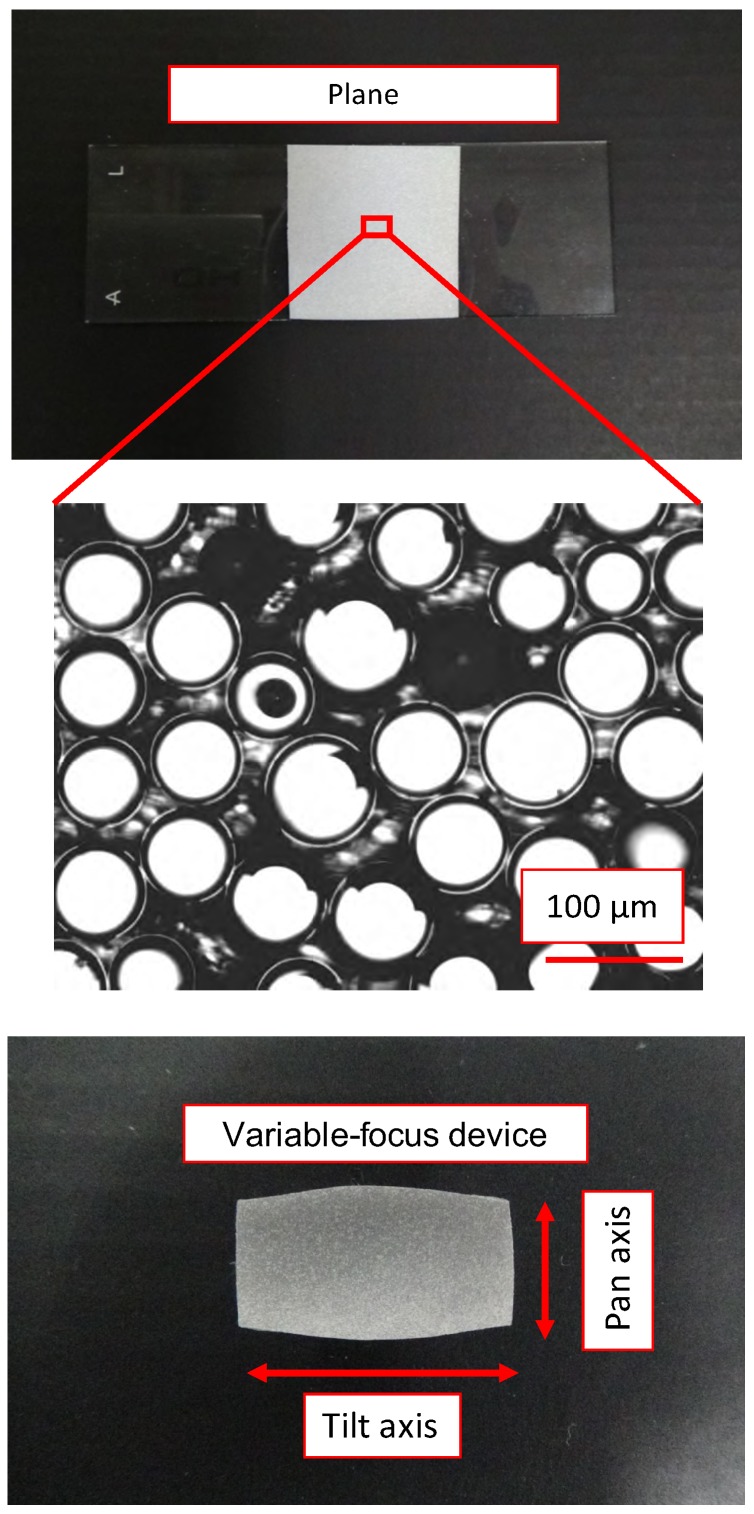
Observation object for evaluation experiments.

**Figure 7 sensors-20-01967-f007:**
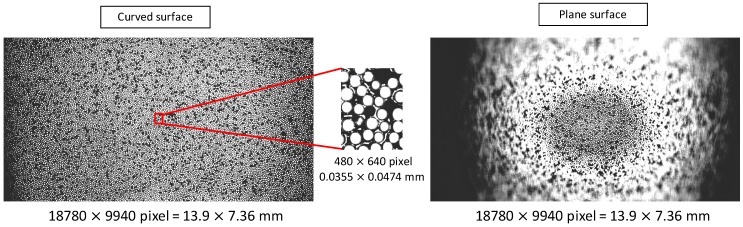
Composite images.

**Figure 8 sensors-20-01967-f008:**
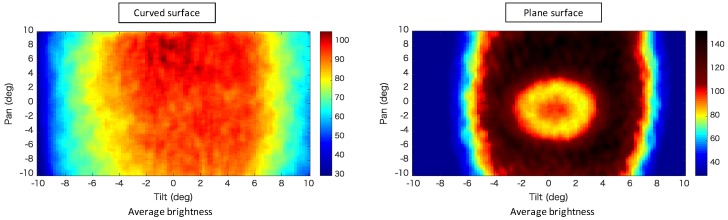
Colormap of the average luminance values with respect to the pan- and tilt-axis mirror angles.

**Figure 9 sensors-20-01967-f009:**
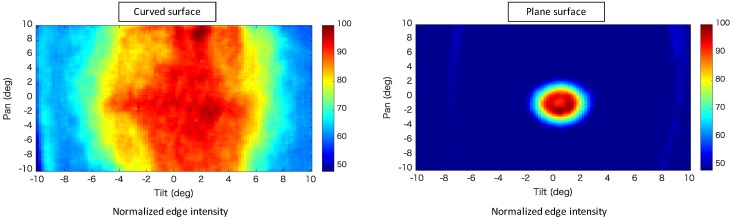
Colormap of the average normalized edge intensity values with respect to the pan- and tilt-axis mirror angles.

**Figure 10 sensors-20-01967-f010:**
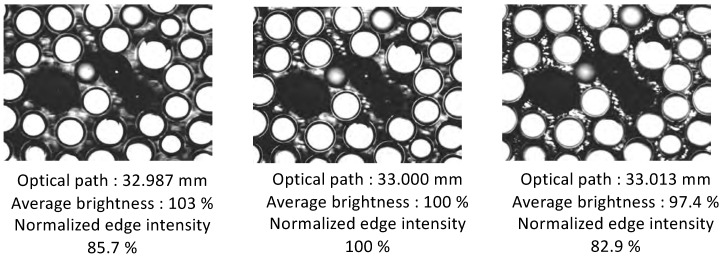
Images with edge intensities is within the threshold value.

**Figure 11 sensors-20-01967-f011:**
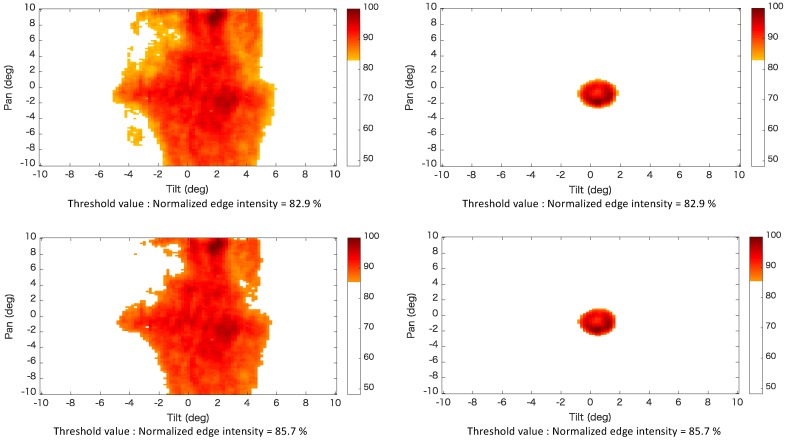
Color map of the observable area that satisfies the threshold values of normalized edge intensity.

**Figure 12 sensors-20-01967-f012:**
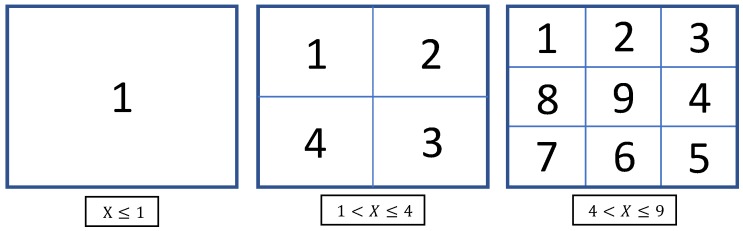
Order in which images were acquired.

**Figure 13 sensors-20-01967-f013:**
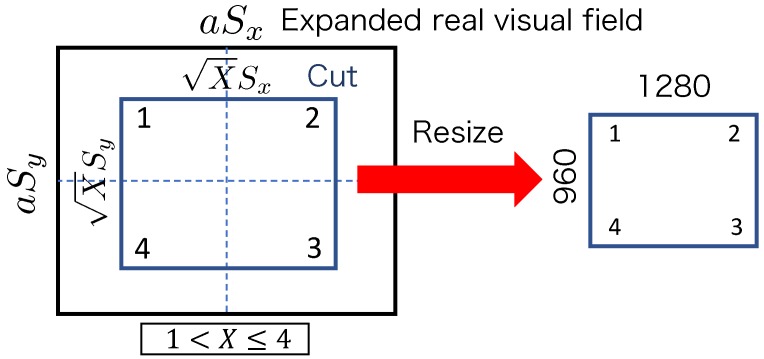
Resizing procedure for HMD.

**Figure 14 sensors-20-01967-f014:**
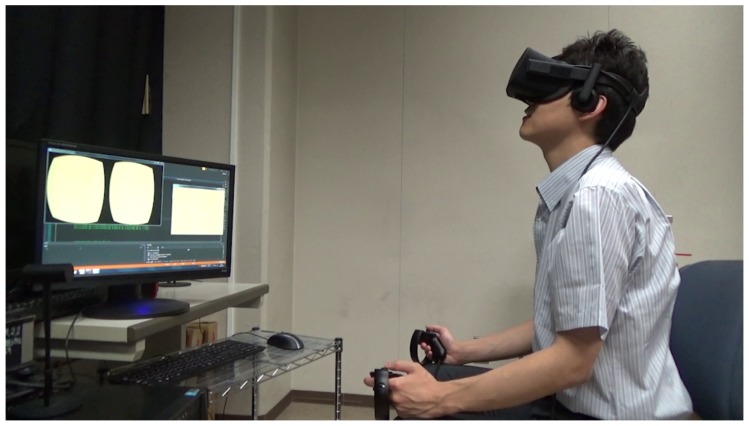
Experimental scene.

**Figure 15 sensors-20-01967-f015:**
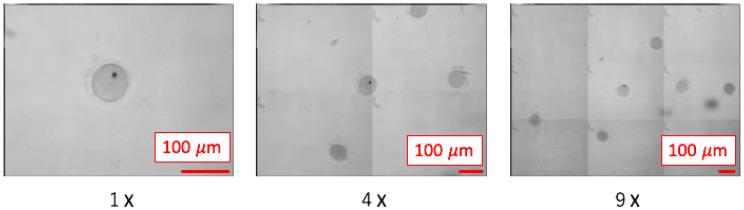
Images that were acquired when the visual field enlargement ratio was set to ×1, ×4, and ×9.

**Table 1 sensors-20-01967-t001:** Comparison of observable areas.

Method of Observing	Observable Area	Rate of Observable Area
Normal	0.144 mm${}^{2}$	1
Plane ($E_{ave}$ = 82.9)	3.017 mm${}^{2}$	21.0
Plane ($E_{ave}$ = 85.7)	2.714 mm${}^{2}$	18.8
variable-focus device ($E_{ave}$ = 82.9)	45.161 mm${}^{2}$	313.6
variable-focus device ($E_{ave}$ = 85.7)	37.645 mm${}^{2}$	261.4

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
