# Peer review of "Head-Mounted Display-Based Microscopic Imaging System with Customizable Field Size and Viewpoint†"

_sensors, 2020, doi:10.3390/s20071967_

Round 1

Reviewer 1 Report

It is an interesting idea to use expansive view to improve visualization via virtual techniques, for microinjection.

Though the authors present the vital results to convey their point, it would be great, if they have used another or typical microinjection model such as mouse embryo instead of Paramesium. 

Also instead of just cartoon of cells at the holding pipette and the injection pipette, if they authors could show a comparison views of normal and expansive views of the actual samples (but also keeping the cartoon as a support-Fig. 1), that would enhance the understanding of their method much better for readers not familiar with the mathematical reasoning and principle behind the expansive view microscopy.

Also the reasoning behind using low magnification objective in the process should be given, as increasing the optical magnification with long working distance objectives would inherently improve the visualization while performing microinjections.

Author Response

Thank you very much for your careful assessment of our paper. We appreciate the opportunity to resubmit a revised paper after implementing the helpful comments and suggestions received from the reviewer. The modifications in the revised paper are highlighted in red font. The specific responses to the reviewer are in the attached file.

Reviewer 2 Report

The paper demonstrated a microscopic imaging system with customized field size and viewpoint. The imaging system enlarges the field view and achieves a high resolution. Observations of Paramecium were carried out by using the developed system. I think this is a useful method worth to be published in Sensors. The manuscript is clearly written and the presented method is well supported by experimental measurements. I recommend publication in Sensors after the following revisions.

  1. Please add the description of the response time of the imaging system.
  2. According to Rayleigh Criterion, the light wavelength and numerical aperture of the optical element have an effect on the resolution of the imaging system. Please add some explanations about the designed system.

Author Response

(The authors gave the same response as above.)

Reviewer 3 Report

The submitted article by Aoyama et al. describes the development of an optical viewing and imaging system, which allows to observe microscopic objects with a head mounted display in real time. The viewpoint of this system is adjusted by input to the virtual reality system, which controls the angle of two Galvano mirrors in the optical path. The implementation of a view-expansive system, combined with a variable focus device, has a greatly expanded field of view and better optical properties compared to conventional view-expansive microscopic systems, which was demonstrated by the imaging of retroreflective sheets.

Broad comments:

  • Figure legends should contain detailed information; the reader should not have to refer to the body text to understand the figures. In turn, the body text can be decluttered by putting the relevant details in the legends.
  • The authors place a lot of emphasis on the use of this optical system for microinjections. However, the actual feasibility for microinjections is not demonstrated nor discussed. For instance, is the curved surface of the variable focus device suited for this purpose, etc.? See also comments below.
  • The evaluation of observable area relies on imaging of reflective sheet, which presumably generates more signal intensity compared to objects that will be used in microinjection. How transferrable are the results from the reflective sheet to a setup with biological samples? It might be useful to compare images of the biological sample at different stage positions within the observable area.
  • Live (“real time”) view via head mounted display seems to be an important feature for a successful microinjection setup. How many frames per second can be achieved and displayed per different resolution/magnification/FOV settings in live view, and what is the corresponding lag time?

Specific comments:

  • The body text itself is mostly well written, and contains few typos or grammatical errors that should be corrected, e.g. lines 98, 107.
  • Lines 12, 216. It is mentioned that the “observable area is 261.4 and 13.9 times larger… “, however, these numbers do not align with the numbers in table 1?
  • In Fig. 3 the mirrors appear out of focus and the lighting is not ideal to discern the installation.
  • Lines 153, 154. Presumably the values were “set to” 100%? Further, this information belongs in the figures’ legends.
  • Line 207. “… the operator did not touch the microscope with the HMD…”?

Author Response

(The authors gave the same response as above.)

Round 2

Reviewer 1 Report

Modification and additional data requested on suggested edits have been performed by authors.